# Microstructure, Mechanical Properties and Oxidation Resistance of Nb-Si Based Ultrahigh-Temperature Alloys Prepared by Hot Press Sintering

**DOI:** 10.3390/ma16103809

**Published:** 2023-05-18

**Authors:** Lijing Zhang, Ping Guan, Xiping Guo

**Affiliations:** 1State Key Laboratory of Solidification Processing, Northwestern Polytechnical University, Xi’an 710072, China; 2School of Chemistry and Chemical Engineering, Northwestern Polytechnical University, Xi’an 710129, China

**Keywords:** Nb-Si based ultrahigh-temperature alloys, hot press sintering, microstructure, room temperature fracture toughness, hardness, oxidation resistance

## Abstract

Nb-Si based ultrahigh-temperature alloys with the composition of Nb-22Ti-15Si-5Cr-3Al (atomic percentage, at. %) were prepared by hot press sintering (HPS) at 1250, 1350, 1400, 1450 and 1500 °C. The effects of HPS temperatures on the microstructure, room temperature fracture toughness, hardness and isothermal oxidation behavior of the alloys were investigated. The results showed that the microstructures of the alloys prepared by HPS at different temperatures were composed of Nbss, βTiss and γ(Nb,X)_5_Si_3_ phases. When the HPS temperature was 1450 °C, the microstructure was fine and nearly equiaxed. When the HPS temperature was lower than 1450 °C, the supersaturated Nbss with insufficient diffusion reaction still existed. When the HPS temperature exceeded 1450 °C, the microstructure coarsened obviously. Both the room temperature fracture toughness and Vickers hardness of the alloys prepared by HPS at 1450 °C were the highest. The alloy prepared by HPS at 1450 °C exhibited the lowest mass gain upon oxidation at 1250 °C for 20 h. The oxide film was mainly composed of Nb_2_O_5_, TiNb_2_O_7_, TiO_2_ and a small amount of amorphous silicate. The formation mechanism of oxide film is concluded as follows: TiO_2_ forms by the preferential reaction of βTiss and O in the alloy; after that, a stable oxide film composed of TiO_2_ and Nb_2_O_5_ forms; then, TiNb_2_O_7_ is formed by the reaction of TiO_2_ and Nb_2_O_5_.

## 1. Introduction

The next-generation aero-engines with high thrust-to-weight ratio have put forward requirements on the temperature-enduring capacity of their hot-end components [1,2]. The operating temperature of the most advanced Ni-based single crystal superalloy is about 1150 °C, which is basically close to their limit (85% of their melting temperatures) [3]. Therefore, it has become an urgent task to develop new high-temperature metallic structural materials that can serve for a long time at temperatures above 1200 °C. As Nb-Si based ultrahigh-temperature alloys with additions of alloying elements such as Al and Cr, etc., have a high melting point (above 1750 °C), relatively low density (6.6–7.2 g/cm^3^), good high temperature creep strength and good fatigue behavior, they have been considered as a prospective high-temperature structural material and have attracted much attention in recent years [4,5,6,7,8].

At present, non-consumable arc melting or directional solidification are employed frequently to prepare Nb-Si based ultrahigh-temperature alloys, which possess microstructures consisting of Nb solid solution (Nbss) and silicides ((Nb,X)_5_Si_3_ (“X” represents the elements substituting for Nb such as Ti and Cr)) [9,10,11,12,13,14]. However, there exist some drawbacks such as serious compositional segregation, coarse microstructure and distinct microstructural heterogeneity in Nb-Si based alloy ingots produced by the aforementioned solidification methods, which would result in poorer mechanical properties. Therefore, seeking a method to prepare Nb-Si based ultrahigh-temperature alloys with uniform composition, even phase distributions and fine microstructure is of great importance. Powder metallurgy can synthesize products with fine microstructure at the micrometer level, which shows excellent high-temperature plastic deformation ability and even superplasticity [15,16,17]. Due to their extremely high melting temperature, it is reasonable to prepare this kind of ultrahigh-temperature alloy by powder metallurgy. The powder metallurgy route generally consists of two procedures, i.e., first preparing the powder blends by mechanical alloying and then sintering the powder blends using methods such as HPS [18,19]. A few researchers have fabricated Nb-Si based alloys by powder metallurgy methods [20,21,22]. However, only the microstructure and mechanical properties of Nb-Si binary and Nb-Si-Fe or Nb-Si-Ti ternary alloys have been studied [20,21,22]. Until now, there is no report in the open literature about the preparation, microstructural characteristics and properties of more practical multicomponent Nb-Si based ultrahigh-temperature alloys through a powder metallurgy route.

Based on the previous research results [10,13,14,23,24,25], optimized Nb-Si based ultrahigh-temperature alloy powders with a composition of Nb-22Ti-15Si-5Cr-3Al (hereafter, all compositions are given in at. % unless otherwise stated) will be fabricated by powder metallurgy. Temperature is an important parameter in the HPS process. Sintering is a temperature-controlled thermal activation process. As the temperature increases, the temperature-related rate factor (e.g., diffusion rate) in the sintering process increases, resulting in an increase in densification rate [26]. To our best knowledge, up to now, there is no systematic research on the effect of HPS temperature on the microstructure and properties of Nb-Si based ultrahigh-temperature alloys. In the present work, the microstructure, room temperature fracture toughness, microhardness and oxidation behavior at 1250 °C of Nb-22Ti-15Si-5Cr-3Al ultrahigh-temperature alloys prepared by HPS at different temperatures have been integrally evaluated.

## 2. Experimental Procedures

Nb, Ti, Si, Cr and Al elemental powders were weighed according to the composition of Nb-22Ti-15Si-5Cr-3Al. Mechanical alloying was performed for 20 h in a planetary ball mill using stainless steel jars and milling balls under Ar atmosphere. The ball milling speed and ball-to-powder ratio (BPR) were fixed at 400 rpm and 15:1. A total 1.25 wt.% stearic acid was added as the process control agent. Block Nb-Si based alloys were prepared in a HVHS-2 ultrahigh-temperature and high-vacuum HPS furnace under Ar atmosphere. The furnace was designed and produced by SKY technology development Co., Ltd. (Shenyang, China), Chinese Academy of Sciences; it uses U-shaped tungsten rods as the heating unit and WRW5Z-395 tungsten rhenium thermocouple as the temperature measurement and control device, with a temperature control accuracy of ±1 °C. The evaluated HPS temperatures were 1250, 1350, 1400, 1450 and 1500 °C.

X-ray diffraction analysis (XRD, X’Pert Pro, Cu *K_α_*, PANalytical, Almelo, The Netherlands) was used to identify the constituent phases in the specimens. The microstructure of the specimens was observed by scanning electron microscopy (SEM, MIRA3 XMU, TESCAN, Brno, Czech Republic) and the chemical compositions of their constituent phases were analyzed by energy dispersive spectroscopy (EDS, Inca X-sight, OXFORD Limited, Oxford, United Kingdom). The refined microstructure of the specimens was examined by a Tecnai F30 G2 transmission electron microscope (TEM) equipped with an EELS system (Gatan GIF Tridiem 863, AMETEK, Berwyn, PA, USA) operating at 300 kV. The Image-Pro Plus 7.0 software (IPP 7.0) was used for quantitative analysis of the volume fraction of constituent phases and the area fraction of pores in the alloys. To improve the statistical accuracy, five BSE images with the same magnification (×1000) were used for each alloy.

The single edge notched three-point-bending specimen for room temperature fracture toughness (*K_Q_*) measurement with dimensions of 25 mm × 5 mm × 2.5 mm was prepared by electro-discharge machining (EDM), and the notch is within the depth of 2.5 ± 0.1 mm. The *K_Q_* value was measured on an electronic universal mechanical testing machine (Instron-3382, INSTRON Corporation, High Wycombe, United Kingdom). Three specimens were tested and the average *K_Q_* value was employed for each specimen condition.

The density of the specimens were measured using Archimedes’ drainage method, based on which the relative density of each specimen was calculated [27]. The Vickers hardness (HV) of the specimens was measured using an HMV-2T (Shimadzu, Hitachi Corporation, Japan) hardness machine with a load of 0.98 N. Each employed hardness value was the average of at least 5 indentations.

The specimens used for oxidation tests were 7 mm × 7 mm × 7 mm cubes cut from the sintered plates, 30 mm in diameter and 10 mm in thickness, by EDM. After the six surfaces were ground, the actual dimensions of each specimen were measured using a vernier caliper and its surface area was calculated before oxidation tests. The isothermal oxidation behavior of the specimens was examined in an open-ended tube furnace. After holding them at 1250 °C for 20 h, the specimens were furnace-cooled down to room temperature. The mass changes of the specimens after oxidation tests were obtained using an electronic balance with an accuracy of 0.001 g.

## 3. Results and Discussion

### 3.1. Phase Analysis of the Alloys

In order to determine the phase constituents of the Nb-Si based ultrahigh-temperature alloys prepared by HPS at different temperatures, XRD analysis was performed on each alloy prepared, as shown in Figure 1. It can be seen that the alloy has diffraction peaks of three phases, which are calibrated as Nbss, γ(Nb,X)_5_Si_3_ (“X” represents Ti, Cr and Al) and βTiss (determination of the diffraction peak of βTiss will be described in the subsequent sections). The γ(Nb,X)_5_Si_3_ intermetallic compound that appeared was generated by the in situ reaction during HPS. At the same time, no diffraction peaks of Cr, Si and other components were found in the Nb-Si based ultrahigh-temperature alloy’s bulk samples prepared by HPS, indicating that Cr, Si and other components all participated in the reaction during HPS: on one hand, they dissolved into the Nb lattice to form Nbss; on the other hand, the γ(Nb,X)_5_Si_3_ silicide was generated due to the chemical reaction among them and other components. In addition, the intensity of the diffraction peak of the γ(Nb,X)_5_Si_3_ intermetallic compound in the Nb-Si based ultrahigh-temperature alloys prepared by HPS at 1250 °C was relatively weak, and the Nbss diffraction peak is still the main peak in the patterns. However, when the HPS temperature was higher than 1250 °C, compared with the diffraction peaks of Nbss, those of γ(Nb,X)_5_Si_3_ were obviously enhanced, which indicates that an increase in HPS temperature promotes the formation of intermetallic compounds through the reaction among Nb, Ti and Si in the milled powders. The in situ reaction in the milled powders began to occur when the HPS temperature was higher than 1250 °C.

### 3.2. Microstructure of the Alloys

Figure 2 shows the backscattered electron (BSE) images of the microstructure of the Nb-Si based ultrahigh-temperature alloys prepared by HPS at different temperatures. It can be seen that the alloy consists of three different contrast areas: white, gray and dark gray. Combined with XRD and EDS analysis results, it can be concluded that the white region is Nbss, the gray one is γ(Nb,X)_5_Si_3_ and the dark gray one is βTiss.

As can be seen in Figure 2a, when the HPS temperature was 1250 °C, the microstructure of the alloy showed obvious river-like strip morphologies, with the length of the strip structure reaching tens of microns and the width only being about a few microns, wherein the Nbss strips are basically parallel to each other. When HPS was carried out at 1250 °C, the powder particles softened under the simultaneous application of pressure such that the lamellar microstructure tended to be distributed perpendicular to the direction of press stress, thus forming a river-like microstructure morphology parallel to each other.

As can be seen in the enlarged view in the upper right corner of Figure 2a, γ(Nb,X)_5_Si_3_ and βTiss were basically distributed in the interface area of Nbss. From this, it can be inferred that γ(Nb,X)_5_Si_3_ and βTiss were precipitated from the decomposition of the supersaturated Nbss in ball-milled powder particles during the HPS process. Table 1 shows the EDS composition analysis results of different phases in Nb-Si based ultrahigh-temperature alloys prepared by HPS at 1250 °C. It can be seen that the content of solid solution elements in Nbss is still very high (especially Si); so, it is still supersaturated Nbss. The gray region is γ(Nb,X)_5_Si_3_ and the black region is βTiss, which indicates that the HPS temperature of 1250 °C is too low; therefore, the milled powder particles do not fully react and obvious diffusion does not occur, resulting in a large amount of supersaturated Nbss strips still existing in the sintered alloy bulk.

Figure 2b,c show the microstructures of Nb-Si based ultrahigh-temperature alloys prepared by HPS at 1350 and 1400 °C, respectively. It can be seen that the morphology of each phase in the microstructure of the alloy changed obviously. In most areas of the alloy, the white, gray and dark gray phases changed from irregular shapes in the alloys HPSed at 1250 °C to nearly equiaxed grain shapes, and the phase interfaces evolved to be clearer and more distinguishable. However, there are still large blocks of Nbss in local areas. Table 2 shows EDS composition analysis results of different phases in Nb-Si based ultrahigh-temperature alloys prepared by HPS at 1350 and 1400 °C. It can be seen that the contents of Ti, Si, Cr and other solute elements in the large blocks of Nbss are higher than those in the Nbss located in the homogeneous microstructural areas, indicating that the large blocks of Nbss are still a supersaturated solid solution. As the HPS temperature increases from 1350 to 1400 °C, the number of large Nbss blocks decreases. It is inferred that the elements in the powders can hardly diffuse with each other and react fully when HPS is performed at a temperature below 1400 °C.

Figure 2d,e show the microstructure of the Nb-Si based ultrahigh-temperature alloys prepared by HPS at 1450 and 1500 °C, respectively. It can be seen that the supersaturated Nbss in the microstructure disappeared and the phases are evenly distributed. The white, gray and dark gray phases completely changed from irregular shapes when the HPS temperature was 1250 °C to nearly equiaxed grain morphologies in the alloys HPSed at a temperature above 1450 °C, with clear and distinguishable phase interfaces, fine microstructures and the size of each grain being smaller than 5 µm. From this phenomenon, it can be concluded that the higher HPS temperature provides a greater driving force for the in situ formation of intermetallic compounds; at the same time, recrystallization is easier to occur at a higher HPS temperature, such that the morphology of each phase grain in the microstructure tends to be regular and their distribution is more uniform. When the HPS temperature continues to increase to 1500 °C, the phase grains in the microstructure of the alloy are still equiaxed, and the morphology and distribution of the constituent phases are basically the same as those prepared by HPS at 1450 °C. However, it can be clearly observed that the microstructure of the alloy prepared by HPS at 1500 °C is significantly coarsened and some silicides have dimensions larger than 5 μm.

From the microstructure of Nb-Si based ultrahigh-temperature alloys prepared by HPS at different temperatures, it was found that the alloy has three phases with different contrasts. Combined with EDS composition analysis, it can be determined that the corresponding three phases are Nbss, γ(Nb,X)_5_Si_3_ and βTiss. However, only the diffraction peaks of two phases, namely, Nbss and γ(Nb,X)_5_Si_3_, are shown in the corresponding XRD patterns. In order to further determine the crystal structure of Tiss, the microstructure of the Nb-Si based ultrahigh-temperature alloys prepared by HPS at 1450 °C was analyzed by TEM, as shown in Figure 3. Figure 3a is a bright field image of the microstructure, and Figure 3b,c are the selected area electron diffraction (SAED) patterns of the corresponding regions. It can be seen that Nbss and Tiss exist in the microstructure, and the crystal structure of Tiss is body-centered cubic (bcc), indicating that it is beta-type. Since the β-type Tiss and Nbss have the same crystal structure and their lattice constants are very close to each other, the diffraction peaks of Tiss and Nbss are almost overlapped in XRD patterns, which is why there are only two sets of diffraction peaks in the XRD patterns. During HPS process, the supersaturated Nbss in the ball-milled powders reacts in situ to form the equilibrium Nbss, γ(Nb,X)_5_Si_3_ and βTiss phases.

Figure 3d is a plane distribution diagram of elements corresponding to TEM bright field images of samples prepared by HPS at 1450 °C. It can be clearly seen that there are voids at the phase interface. Therefore, in addition to the large black region in the microstructure (Figure 2), which is a hole, the small black area at the interface between different phases should also be a hole. However, the reasons for the formation of these two holes are different. The former may be caused by the release of residual gas between powder particles during HPS, while the latter may be caused by the difference in the molar volume of different phases during HPS. As can be seen in Figure 2, as the HPS temperature increases, the porosity in the sintered alloy becomes less. This is due to the increase in recrystallization degree and grain growth of the alloy with the increase in sintering temperature. Quantitative metallography analysis by IPP 7.0 shows that the area fraction of the pores in the HPSed alloys decreases with the increase in HPS temperature from 1.2% at the HPS temperature of 1250 °C to 0.8% at 1350 °C, 0.7% at 1400 °C, 0.4% at 1450 °C and 0.4% at 1500 °C. The area fractions of porosities are the same when the HPS temperature is 1450 °C and 1500 °C.

### 3.3. Physical Characters of the Alloys

Figure 4 shows the density and relative density changes of Nb-Si based ultrahigh-temperature alloys prepared by HPS at 1250, 1350, 1400, 1450 and 1500 °C, measured by Archimedes drainage method. When sintered at 1250 °C, the density of the alloy is only 6.593 g/cm^3^; thus, the densification degree is relatively low. The density of the alloy increases rapidly with the increase in sintering temperature. However, when the HPS temperature is higher than 1400 °C, the increase in density becomes lower, especially after 1450 °C. When the HPS temperature was 1450 °C, the density of the alloy reached 6.905 g/cm^3^. The density of the alloy was still 6.907 g/cm^3^ when the HPS temperature was increased to 1500 °C, which is nearly the same as that of the alloy sintered at 1450 °C. The variation of the relative density of the alloys also presents an obvious increasing tendency with the increase in HPS temperature, which is consistent with the variation in the alloy’s density [27]. The density of the sintered alloys is closely related to the in situ reaction during the sintering process. Increasing the sintering temperature can promote the in situ reaction in the milled powders to form intermetallic compounds, as has been proved by the XRD patterns in Figure 1 and microstructure in Figure 2. Therefore, a higher HPS temperature can lead to higher density and relative density of the alloys. However, when the HPS temperature is increased to 1450 °C, sufficient atomic diffusion and reaction can already occur among the as-milled alloy powders; thus, further increasing the HPS temperature no longer obviously increases the density and relative density of the alloy.

Figure 5 shows the average size changes of the Nbss, γ(Nb,X)_5_Si_3_ and βTiss phase grains in Nb-Si based ultrahigh-temperature alloys prepared by HPS at different temperatures. It can be seen that the Nbss grain size first decreases and then increases with the increase in HPS temperature, and its size is the smallest when the HPS temperature is 1450 °C. This is because the supersaturated Nbss in the ball-milled powders still remains in the alloy sintered at temperatures below 1450 °C. However, when the HPS temperature is higher than 1450 °C, the in situ reaction of the powders and even the recrystallization ability of the alloys are enhanced, resulting in the recrystallization of Nbss grains and the obvious occurrence of Nbss grain growth. Differently from Nbss, the sizes of γ(Nb,X)_5_Si_3_ and βTiss phase grains increase with the increase in HPS temperature, because both phases are in situ precipitated from supersaturated Nbss powders during the HPS process and are essentially not affected by the size of the original powders. Therefore, with the increase in HPS temperature, the precipitation rate of both γ(Nb,X)_5_Si_3_ and βTiss becomes faster and their grain size increases continuously. The size of these two phases increased obviously in the microstructure of the alloy prepared by HPS at 1500 °C, which may be due to the exponential relationship between recrystallization rate and temperature. On the whole, the microstructure of Nb-Si based ultrahigh-temperature alloys prepared by HPS is significantly finer than that of the arc melted alloy, since the size of Nb_5_Si_3_ in the arc melted alloy is generally in the tens of microns [10].

Figure 6 shows the volume fraction of each phase in the microstructure of Nb-Si based ultrahigh-temperature alloys prepared by HPS at different temperatures. It can be seen that when the HPS temperature is lower than 1450 °C, the volume fraction of Nbss phase decreases continuously from 48.77% at 1350 °C to 38.02% at 1450 °C. The volume fraction of γ(Nb,X)_5_Si_3_ phase increased from 38.65% in the alloy sintered at 1350 °C to 44.97% in the alloy sintered at 1450 °C, and that of the βTiss phase from 12.58% to 17.01%, respectively. This further shows that the alloy powders will react more thoroughly with the increase in HPS temperature until 1450 °C. However, when the HPS temperature further increases to 1500 °C, the volume fraction of each phase does not change obviously. This indicates that at 1450 °C, the as-milled Nb-Si based ultrahigh-temperature alloy powders have undergone sufficient atomic diffusion and reaction during the HPS process, and all the final products—namely, Nbss, γ(Nb,X)_5_Si_3_ and βTiss phases—have formed.

### 3.4. Room Temperature Fracture Toughness

Table 3 presents the room temperature fracture toughness *K_Q_* values of the Nb-Si based ultrahigh-temperature alloy prepared by HPS at different temperatures. It can be seen that when the HPS temperature is lower than 1450 °C, the *K_Q_* values of the alloy exhibit an increasing trend with the increase in HPS temperature. This should be a comprehensive improvement, including increased alloy density and more uniform and refined grains with the increase in HPS temperature. However, when the HPS temperature further increased to 1500 °C, the *K_Q_* values of the alloy decreased slightly. This can be attributed to the fact that the density of the alloy does not change significantly when the HPS temperature increases from 1450 °C to 1500 °C but the equiaxed grains coarsen a little.

Figure 7 shows the crack propagation paths of the three-point bending specimens of Nb-Si based ultrahigh-temperature alloys prepared by HPS at 1400 and 1450 °C. It can be seen that the crack in the alloy prepared by HPS at 1400 °C propagates straightly. The crack propagation of the specimen of the alloy was not hindered effectively by the microstructure. This may be due to the low density of the alloy prepared by HPS at 1400 °C, meaning the cracks could propagate rapidly along the fine voids in the microstructure. The crack propagation path of the alloy prepared by HPS at 1450 °C is similar to that of the alloy prepared at 1400 °C. However, crack bridging and crack deflection occurs during the crack propagation process of the specimen prepared by HPS at 1450 °C (Figure 7b), which indicates that the microstructure hinders the crack propagation effectively and, thus, improves the room temperature fracture toughness of the alloy obviously.

Figure 8 shows the SEM images of fracture surfaces of the three-point bending specimens of the Nb-Si based ultrahigh-temperature alloys prepared by HPS at different temperatures. It can be seen that the fracture surfaces of the alloys prepared by HPS at 1350 and 1400 °C are smooth. The density of the alloys prepared at these two temperatures is not high and there are a lot of pores. As a result, the fracture surfaces of the alloys show a honeycomb-like morphology. The existence of pores leads to increased brittleness of the alloys. After HPS at 1450 and 1500 °C, the fracture surfaces of the three-point bending specimens of the alloys show relative fluctuations, indicating that the alloys have undergone a certain degree of plastic deformation. The reason is that the density of the alloys is greatly increased, so there are only a few small-sized pores in the fracture morphology. On the other hand, the Nb-Si based ultrahigh-temperature alloys prepared by HPS at these two temperatures have an equiaxed morphology with a clear interface (as shown in Figure 2). The crack deflects when it propagates to the interface and needs more energy to continue to propagate; so, it has relatively high room temperature fracture toughness values.

### 3.5. Hardness

The Vickers hardness of Nb-Si based ultrahigh-temperature alloys prepared by HPS at different temperatures is listed in Table 4. It can be seen that the Vickers hardness of the alloys gradually increases with the increase in HPS temperature until 1450 °C. This is due to the low density of the alloys and the existence of a large number of voids when the HPS temperature is lower than 1450 °C. The lower the HPS temperature is, the lower the density of the alloy. The existence of pores seriously weakens the ability of the alloys to resist the indentation, which results in the specimen prepared by HPS at lower temperature having a smaller hardness value. As the amount of porosities decreases with the increase in HPS temperature, the Vickers hardness value of the alloys increases gradually. When the HPS temperature rises to 1450 °C, the Vickers hardness value of the alloys is the largest. The Vickers hardness value of the alloys decreases when the HPS temperature is further increased to 1500 °C. This is mainly due to the coarsening of the alloy’s microstructure. The Vickers hardness value of Nb-Si based ultrahigh-temperature alloys prepared by HPS at 1450 °C is about 1138 HV, while the Vickers microhardness of primary silicide in Nb-Si based ultrahigh-temperature alloys prepared by arc melting method is 1083 HV and the microhardness of eutectic structure is only 671 HV [10]. The high hardness value of Nb-Si based ultrahigh-temperature alloys prepared by HPS should be due to grain refinement strengthening caused by the fine grain structure of the HPSed alloys.

### 3.6. Oxidation Behavior of the Alloys

Figure 9 shows the weight gain per unit area of Nb-Si based ultrahigh-temperature alloys prepared by HPS at 1250, 1350, 1400, 1450 and 1500 °C after oxidation at 1250 °C for 20 h. It can be seen that the weight gain per unit area of the alloys decreases significantly with the increase in HPS temperature, from about 62.26 mg/cm^2^ for the alloy prepared by HPS at 1250 °C to about 40.51 mg/cm^2^ for the alloy prepared by HPS at 1450 °C, while the oxidation weight gain per unit area of the alloys HPSed at 1500 °C does not change significantly compared with that of the alloy prepared by HPS at 1450 °C. This indicates that the alloys prepared by HPS at different temperatures have obviously different oxidation performance. When the HPS temperature was 1250 °C, there still remained many microstructural characteristics of the original milled powder particles in the sintered alloys. This indicates that the sintering reaction of the powders is not sufficient at such lower HPS temperature, resulting in the alloy being not dense; thus, its high-temperature oxidation resistance is poor. When the HPS temperature was higher, the sintered alloys had nearly equiaxed grains. The sintering reaction is much more sufficient. The reaction diffusion of each element in the alloys is also sufficient, resulting in a more compact microstructure that is beneficial for forming a denser and more protective oxide scale, thereby improving the high-temperature oxidation resistance of the alloy.

Figure 10 shows the XRD patterns of the oxide films on the Nb-Si based ultrahigh-temperature alloys prepared by HPS at different temperatures after oxidation at 1250 °C for 20 h. Figure 11 shows the cross-sectional BSE morphology of the oxide film. Table 5 presents the EDS analysis results of the composition of the typical phases observed in Figure 11. It can be seen from Figure 11 that the oxide films of the alloy prepared by HPS at different temperatures possess similar microstructures. The XRD patterns show that the oxide films composed of Nb_2_O_5_, TiO_2_ and TiNb_2_O_7_ form on the surface of each alloy after oxidation. EDS analysis of the typical phases in the oxide films shows that the oxygen content in the gray–white microstructure is higher, and the typical components are 9.40Nb-4.53Ti-0.20Al-2.13Cr-0.73Si-83.01O (at. %), as shown by the compositions of points 1, 8 and 12. Combined with the XRD patterns in Figure 10, it can be seen that these gray–white phases are mainly Nb_2_O_5_. The content of Si and O in some black-contrast microstructures existing in the oxide films is higher, as shown by the compositions of points 2 and 9, indicating that it is mainly SiO_2_ or silicate phases. However, it is difficult to find the diffraction peaks of SiO_2_ or silicates in the XRD patterns. The main reason may be that the oxidation products of the silicides exist in amorphous form. Similar XRD patterns of the arc-melted Nb-Si based ultrahigh-temperature alloys after oxidation at 1250 °C were analyzed in our previous research [14]. The atomic ratio of Nb to Ti in gray-contrast microstructure is about 2:1, and the result of XRD analysis on the outer layer of oxide film shows that it is TiNb_2_O_7_. In addition, many fine dark gray phases are dispersed in the outer layer of the oxide film. The Ti content in these dispersed dark gray hues, massive dark gray microstructures with broken surface and continuous dark gray structures at the bottom of the outer layer is high, which are all TiO_2_, combined with XRD analysis results. The reason for the formation of continuous TiO_2_ layer at the bottom of the outer oxide film will be analyzed and discussed later. Figure 11f summarizes the microstructure of the scales formed on the HPSed alloys upon oxidation at 1250 °C for 20 h. It can be seen that the Si-containing oxides or silicate areas and Nb_2_O_5_, TiO_2_ and TiNb_2_O_7_ areas are separately distributed. More TiO_2_ forms at the bottom of the scales.

Figure 12 shows the BSE morphology of the internal oxidation zone of the Nb-Si based ultrahigh-temperature alloys prepared by HPS at different temperatures after oxidation at 1250 °C for 20 h. It can be seen that the internal oxidation zone is composed of three typical phases: black (as shown in points 1, 4, 7 and 10), gray (as shown in points 2, 5, 8 and 11) and light gray (as shown in points 3, 6, 9 and 12). However, there are obvious differences in the microstructural morphologies of the internal oxidation zone of the alloys prepared by HPS at different temperatures. The internal oxidation zone of the alloy prepared by HPS at 1250 °C mainly presents a plate-like microstructure, while that of the alloy prepared by HPS at higher temperature presents nearly a equiaxed grains feature. The EDS analysis results of the compositions of the above typical phases are shown in Table 6. It can be seen that Ti and O contents in the black phase are higher, satisfying the composition of TiO_2_, indicating that it is mainly TiO_2_. The content of Nb and Si in the dark gray phase satisfies the composition of Nb_5_Si_3_ phase, indicating that it is mainly Nb_5_Si_3_ phase in the alloy but there is a certain content of O. However, the content of Nb in the light gray phase is generally higher than 70 at. %, and the rest includes a small content of other elements and a certain content of O, indicating that it is mainly Nbss phase. The results of EDS analysis show that TiO_2_ preferentially formed in the alloys during the oxidation process, which is more closely related to the lower formation energy of TiO_2_ than that of Nb_2_O_5_: the Gibbs free energy of formation of TiO_2_ and Nb_2_O_5_ at 1250 °C is −669.8 kJ and −498.9 kJ, respectively (the reaction value is the standard Gibbs free energy of Ti and Nb when they react with 1 mol of O_2_ to form their corresponding oxides); according to the selective oxidation theory raised by Wagner, oxygen atoms will preferentially react with elements with stronger affinity to generate corresponding oxides when oxygen is limited [28]. In this study, the inward diffusion of oxygen atoms into the metal substrate is hindered by the outer oxide film, creating an environment for the preferential oxidation of Ti in the internal oxidation zone.

## 4. Conclusions

(1)The microstructure of Nb-Si based ultrahigh-temperature alloys prepared by HPS at different temperatures consists of Nbss, βTiss and γ(Nb,X)_5_Si_3_ phases. The microstructure of the alloys prepared by HPS at 1450 °C is fine and nearly equiaxed. However, there are still supersaturated Nbss in the alloys prepared by HPS below this temperature. The microstructure of the alloys obviously coarsens when the HPS temperature is higher than 1450 °C.(2)With the increase in HPS temperature, the density of the prepared Nb-Si based ultrahigh-temperature alloys increases continuously, and the density of the alloy prepared at 1450 °C reaches the maximum. With the increase in HPS temperature, the contents of βTiss and γ(Nb,X)_5_Si_3_ in the alloy gradually increase.(3)The room temperature fracture toughness and Vickers hardness of Nb-Si based ultrahigh-temperature alloys prepared by HPS at 1450 °C are the highest.(4)The oxide films spalled out and there were obvious internal oxidation zones in the alloys prepared by HPS at different temperatures upon oxidation at 1250 °C for 20 h. The oxidation weight gain of the alloy prepared by HPS at 1450 °C is the smallest. The oxide film of the alloy is mainly composed of Nb_2_O_5_, TiNb_2_O_7_, TiO_2_ and a small amount of amorphous silicate.

## Figures and Tables

**Figure 1 materials-16-03809-f001:**
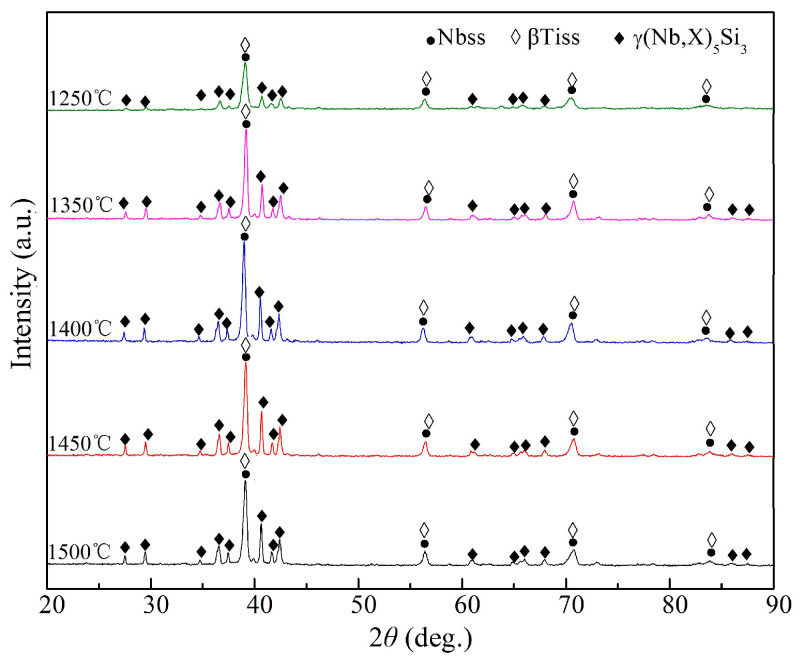
XRD patterns of the Nb-Si based ultrahigh-temperature alloys hot press sintered (HPSed) at different temperatures.

**Figure 2 materials-16-03809-f002:**
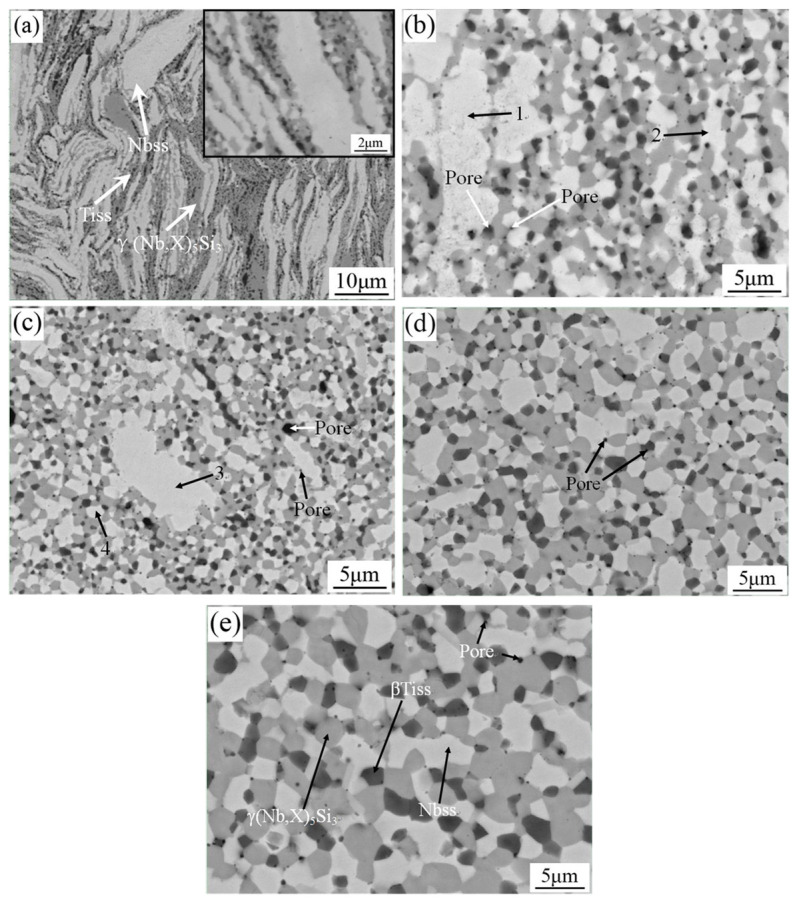
BSE images of the Nb-Si based ultrahigh-temperature alloys HPSed at (**a**) 1250 °C, (**b**) 1350 °C, (**c**) 1400 °C, (**d**) 1450 °C and (**e**) 1500 °C. The arrowhead positions with numerals “1–4” are the locations where EDS analyses were performed.

**Figure 3 materials-16-03809-f003:**
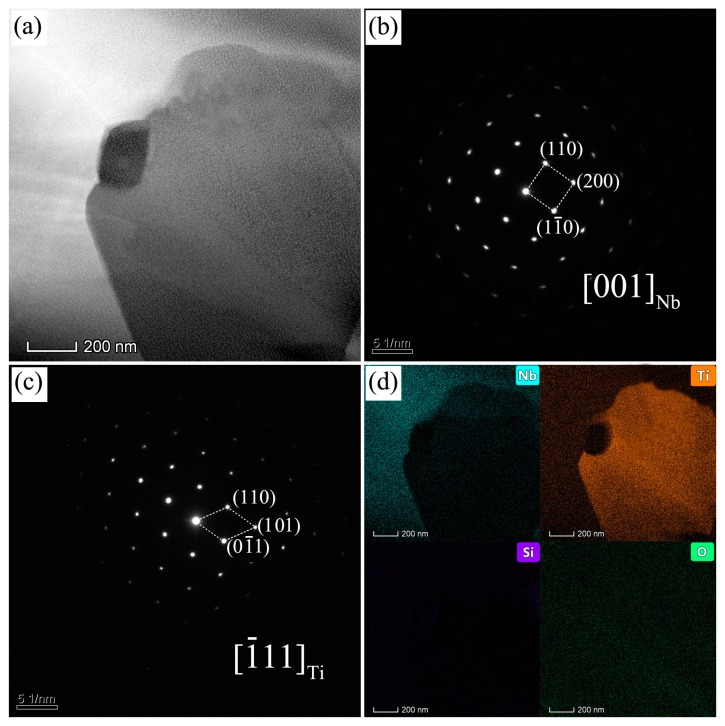
Nb-Si based ultrahigh-temperature alloys HPSed at 1450 °C. (**a**) Bright field image, (**b**) SAED pattern of Nbss, (**c**) SAED pattern of Tiss and (**d**) the element mapping.

**Figure 4 materials-16-03809-f004:**
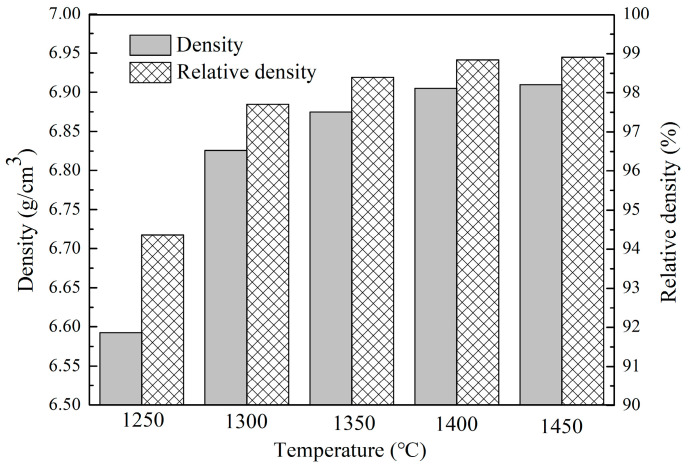
Variation of the density and relative density of the Nb-Si based ultrahigh-temperature alloys vs. HPS temperature.

**Figure 5 materials-16-03809-f005:**
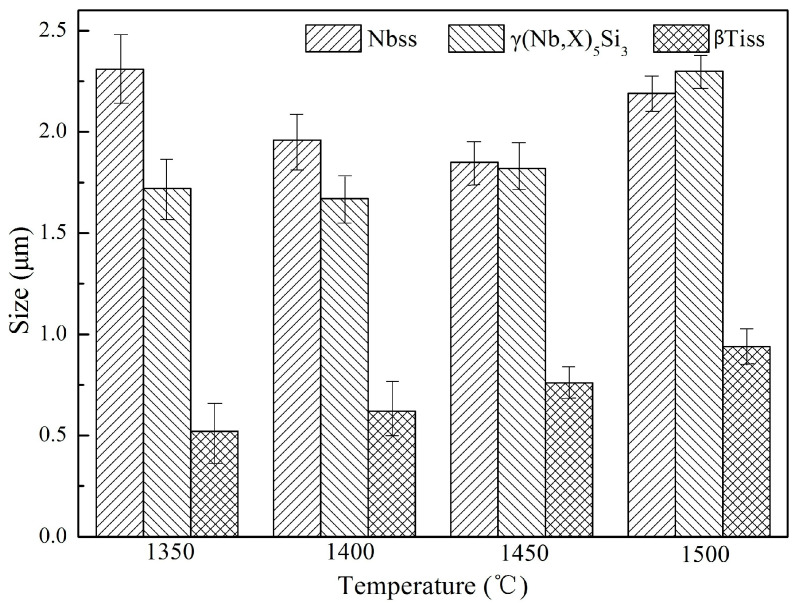
Grain size of the constituent phases in the alloys HPSed at different temperatures.

**Figure 6 materials-16-03809-f006:**
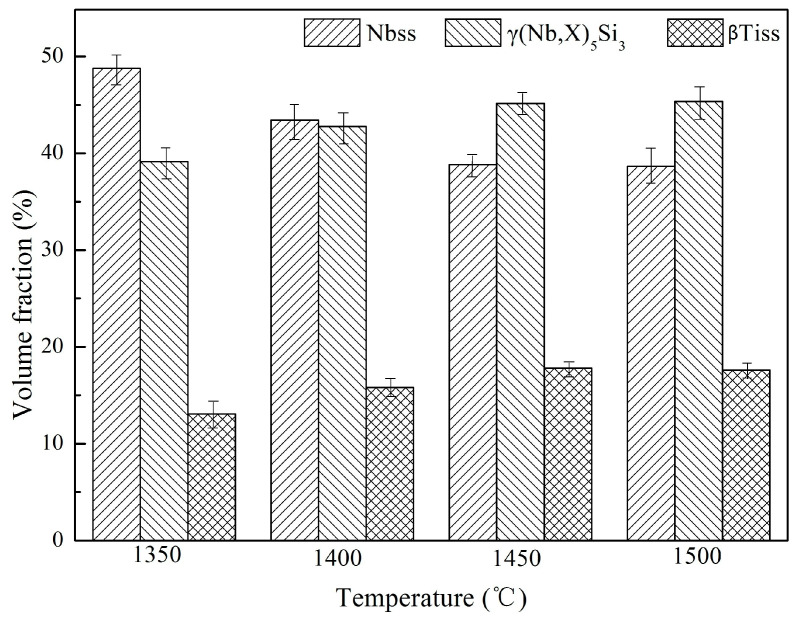
Volume fraction of each phase in the alloys HPSed at different temperatures.

**Figure 7 materials-16-03809-f007:**
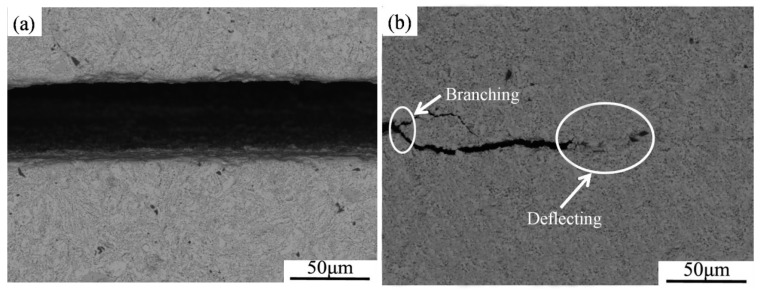
Crack propagation paths of the three-point bending specimens of the Nb-Si based ultrahigh-temperature alloys prepared by HPS at (**a**) 1400 °C and (**b**) 1450 °C.

**Figure 8 materials-16-03809-f008:**
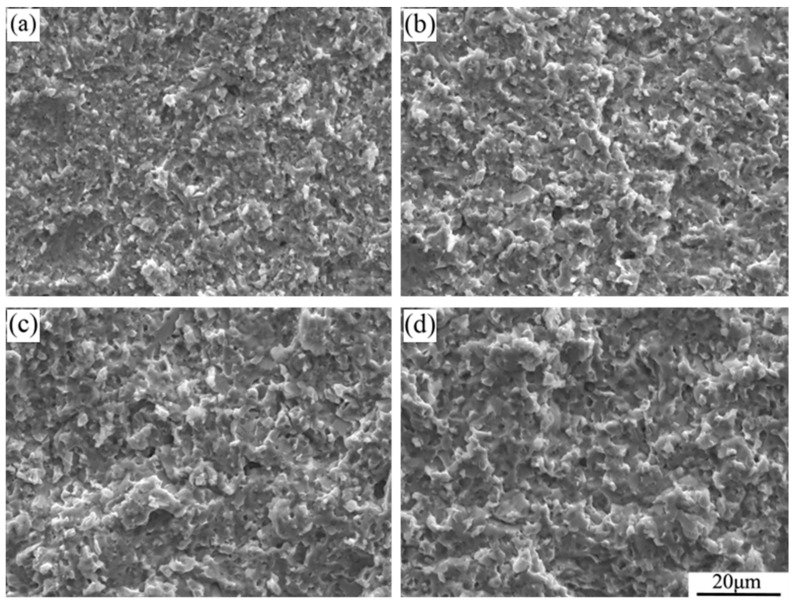
Typical SEM images of the fracture surfaces of the three-point bending specimens of Nb-Si based ultrahigh-temperature alloys prepared by HPS at (**a**) 1350 °C, (**b**) 1400 °C, (**c**) 1450 °C and (**d**) 1500 °C.

**Figure 9 materials-16-03809-f009:**
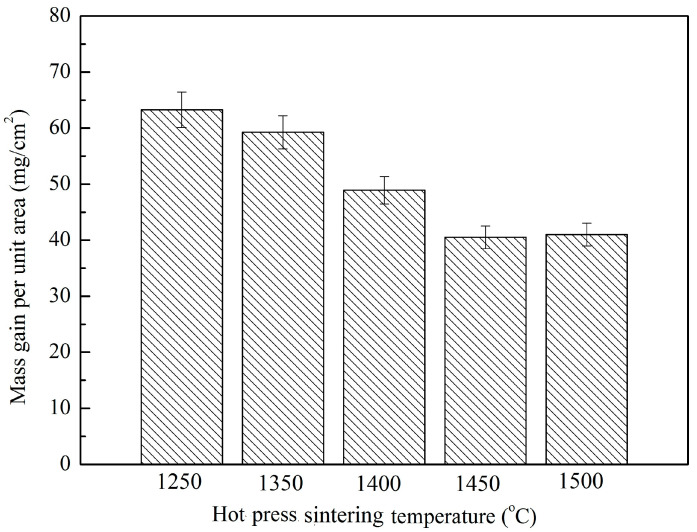
Mass gains per unit area of the Nb-Si based ultrahigh-temperature alloys prepared by HPS at different temperatures upon oxidation at 1250 °C for 20 h.

**Figure 10 materials-16-03809-f010:**
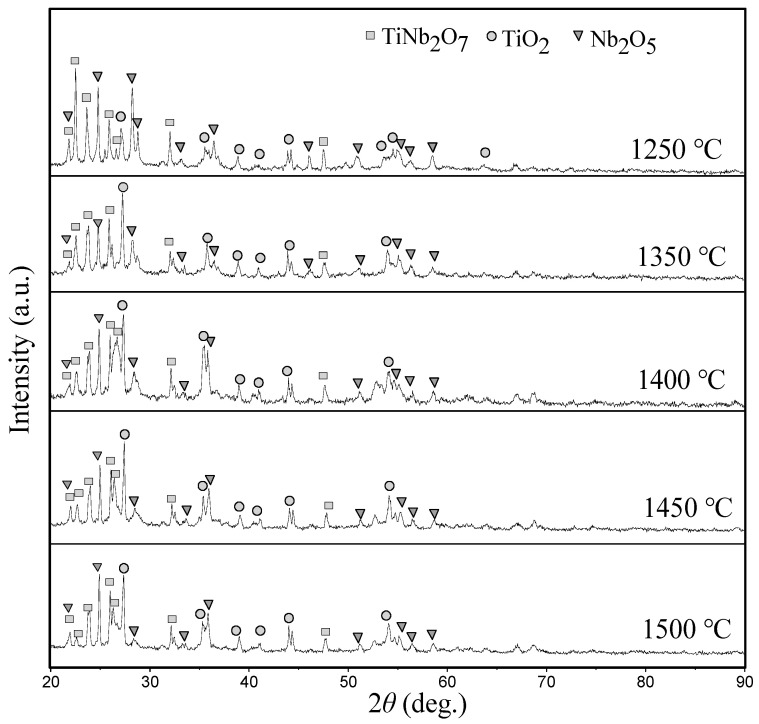
XRD patterns of the outer layers of the scales formed on Nb-Si based ultrahigh-temperature alloys prepared by HPS at different temperature upon oxidation at 1250 °C for 20 h.

**Figure 11 materials-16-03809-f011:**
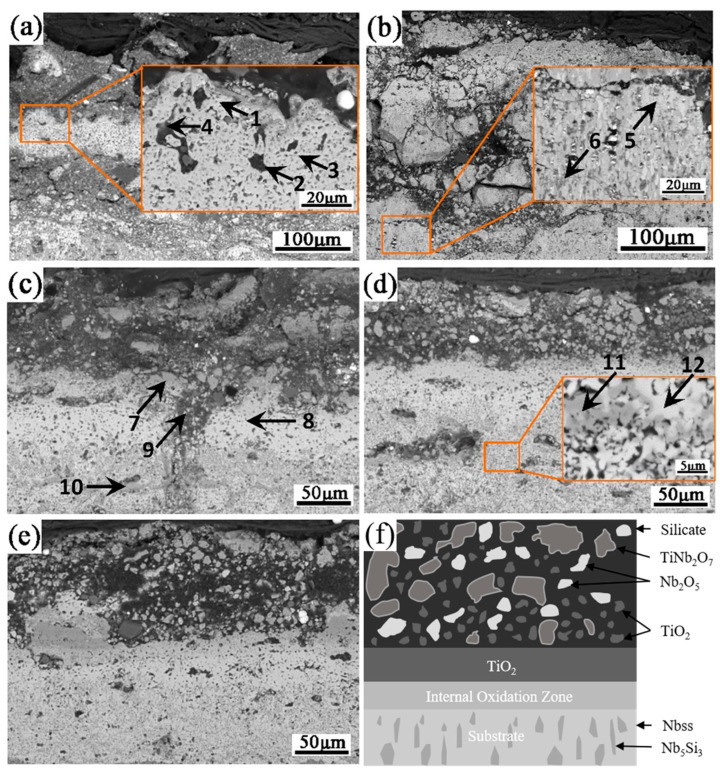
BSE images of the outer layers of the scales formed on Nb-Si based ultrahigh-temperature alloys prepared by HPS at different temperatures upon oxidation at 1250 °C for 20 h. HPSed at (**a**) 1250 °C, (**b**) 1350 °C, (**c**) 1400 °C, (**d**) 1450 °C and (**e**) 1500 °C, and (**f**) is the sketch drawing for the microstructure of the specimen oxidized at 1250 °C for 20 h. The arrowhead positions with numerals “1–12” are the locations where EDS analyses were performed.

**Figure 12 materials-16-03809-f012:**
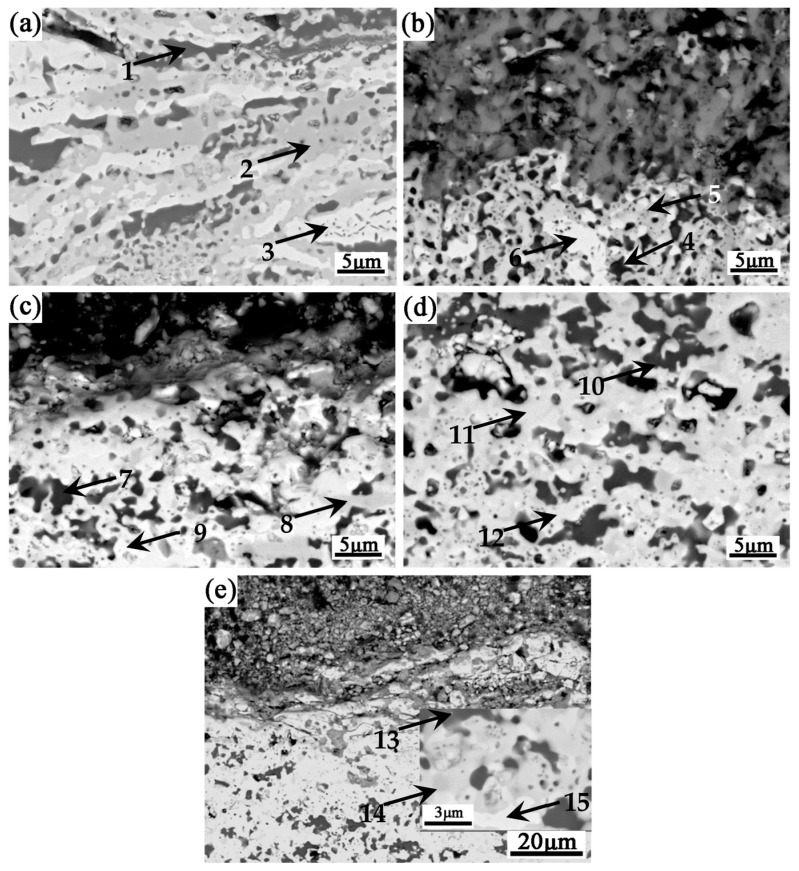
BSE images of the internal oxidation zones of Nb-Si based ultrahigh-temperature alloys prepared by HPS at different temperatures upon oxidation at 1250 °C for 20 h. HPSed at (**a**) 1250 °C, (**b**) 1350 °C, (**c**) 1400 °C, (**d**) 1450 °C and (**e**) 1500 °C. The arrowhead positions with numerals “1–15” are the locations where EDS analyses were performed.

**Table 1 materials-16-03809-t001:** EDS analysis results of each constituent phase in the Nb-Si based ultrahigh-temperature alloys HPSed at 1250 °C shown in Figure 2a, marked by arrowheads.

Constituent Phase	Composition (at. %)				
	Nb	Ti	Si	Cr	Al
Nbss	60.21	17.89	12.80	5.35	3.75
γ(Nb,X)_5_Si_3_	47.01	18.71	33.11	0.36	0.81
βTiss	8.50	86.48	3.62	0.57	0.83

**Table 2 materials-16-03809-t002:** EDS analysis results of the phases marked with numerals “1”, “2”, “3” and “4” in Figure 2b,c.

Sites	Composition (at. %)				
	Nb	Ti	Si	Cr	Al
1	65.96	16.89	8.81	5.25	3.09
2	79.34	8.84	0.93	7.23	3.66
3	68.51	15.33	7.39	5.98	2.79
4	79.98	8.16	0.75	6.91	4.20

**Table 3 materials-16-03809-t003:** Room temperature facture toughness of the Nb-Si based ultrahigh-temperature alloys prepared by HPS at different temperatures.

HPS Temperature (°C)	Average *K_Q_ ±* Error Bar (MPa·m^1/2^)
1350	8.1 ± 0.6
1400	8.7 ± 0.3
1450	10.9 ± 0.4
1500	10.2 ± 0.4

**Table 4 materials-16-03809-t004:** Vickers hardness of Nb-Si based ultrahigh-temperature alloys prepared by HPS at different temperatures.

HPS Temperature (°C)	Vickers Hardness *±* Error Bar (HV)
1350	780 ± 73
1400	885 ± 61
1450	1138 ± 59
1500	979 ± 38

**Table 5 materials-16-03809-t005:** Composition of the constituent phases marked by arrows with numerals “1–12” in Figure 11, determined by EDS analyses.

Sites	Composition (at. %)					
	Nb	Ti	Al	Cr	Si	O
1	21.28	4.45	/	1.22	0.09	72.97
2	10.96	1.49	0.05	0.15	15.53	71.82
3	9.40	4.53	0.20	2.13	0.73	83.01
4	3.15	21.93	0.10	2.51	0.01	72.30
5	18.40	7.18	0.28	0.09	0.72	73.32
6	/	15.13	0.22	/	1.29	83.37
7	7.05	15.18	0.41	6.50	/	70.87
8	22.49	2.77	0.18	0.23	0.25	74.07
9	0.53	0.33	0.06	0.07	33.14	65.87
10	2.32	28.28	/	0.33	3.11	65.96
11	12.20	9.27	0.96	5.83	0.87	70.87
12	20.93	2.73	0.01	0.24	3.05	73.05

**Table 6 materials-16-03809-t006:** Composition of constituent phases marked by arrows with numerals “1–15” in Figure 12, determined by EDS analyses.

Sites	Composition (at. %)					
	Nb	Ti	Al	Cr	Si	O
1	0.16	27.94	0.67	/	0.07	71.16
2	56.38	3.13	0.26	0.44	27.86	11.93
3	75.68	0.30	/	4.22	0.99	18.81
4	8.25	25.59	1.39	0.24	2.61	61.92
5	57.18	3.85	0.30	0.68	21.05	16.95
6	75.05	1.01	/	3.66	0.06	20.22
7	1.60	28.27	0.37	0.20	0.48	69.08
8	54.07	0.45	/	1.11	29.03	15.34
9	72.93	1.86	0.27	3.52	1.30	20.11
10	4.05	28.80	0.55	0.12	0.42	66.06
11	44.71	6.82	0.79	0.41	20.46	26.80
12	78.84	3.27	0.61	0.24	1.38	15.66
13	1.44	29.98	0.32	0.66	0.59	67.01
14	51.17	0.37	/	23.57	3.13	21.75
15	78.64	2.91	2.80	4.95	0.71	14.33

## Data Availability

Further data can be provided upon request.

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
