# Peer review of "Microstructure, Mechanical Properties and Oxidation Resistance of Nb-Si Based Ultrahigh-Temperature Alloys Prepared by Hot Press Sintering"

_materials, 2023, doi:10.3390/ma16103809_

Round 1

Reviewer 1 Report

Manuscript ID: materials-2351020

Dear Editor,

The authors investigated the effect of sintering temperature through HP technique on the microstructure, fracture toughness, hardness and oxidation beahaviour of Nb-Si-Ti-Cr-Al alloy.

1-The English language needs a moderate correction. This both in linguistic correction and difficulty in understanding several statements. Certified English correction is necessary in this matter.

2-Does (at.%) means atomic percentage? If does so, it should be mentioned from the abstract (or at least from beginning in the Introduction section).

3-Is HVHS-II is a kind of furnace? It should be clarified in the experimental section writing the full description of the device, such  as manufacturer and model.

4-The same requirement for point 3 should be rearranged  by the authors for XRD, SEM, EDS, TEM, HV, KQ and other devices' measurements. These corrections should be embedded in the Experimental Procedure Section.

5-in the Results and Discussion section, section 3.1 should reorganized as phase analysis for XRD results and add section 3.2 for microstructure of alloys based on SEM results.

6-for the EDS test, the location of the EDS test should be identified on the SEM micrograph.

7-from Figure 2, SEM micrographs shows pores in the microstructure. However, the authors have nit discussed the pore percentage change with increasing of HP sintering temperature.

8-line 208: Body Centered Cube (BCC)....

9-for density explanation, the authors can add the sub-section of physical properties.

10- The authors described the change in density in figure 4, however, no explanation was given for this change??

11-figure 5 caption should be modified as grain size for ....

12-Figure 6, volume fraction of each phases were shown. In the experimental procedure section, the authors did not describe the methodology to find volume for each phase?

13-for Table 3, the error bar for fracture toughness values is missed.

14-line 299: 1350 C should be changed to 1450 C!

15-Figure 7, to evaluate the difference between two micrographs at approximate temperatures 1440 and 1450 C,

(a) the same scale for each micrograph should be taken.

(b) the density variation between these two temperatures are not significant. Hence, the authors should justify what the reasons (for example porosity variation which mentioned in figure 8 and/or may be other reasons) behind this fracture toughness improvement from 8.7 Mpa.m1/2 to 10.9 MPa.m1/2

16-HV instead of Hv throughout the manuscript.

17- table 4, error bar for HV values is missed.

18-The reference style should be formatted in accordance with Materials journal style citation.

Manuscript ID: materials-2351020

Dear Editor,

The authors investigated the effect of sintering temperature through HP technique on the microstructure, fracture toughness, hardness and oxidation beahaviour of Nb-Si-Ti-Cr-Al alloy.

1-Line 15: ...the obtained microstructure is fine....

2-hot press used too much throughout the text. Just using HP instead can serve the same purpose more efficiently.

3-Line 44-45: However there exist some drawbacks ....should be corrected.

4-Line 46-47: produced by above mentioned solidification ... can replaced with: produced by the aforementioned solidification ....

There are several similar phrases, statements, linguistic and grammatical mistakes should be corrected throughout the submitted manuscript and/or make modifications to  ensure more interesting readability by the readers.

Reviewer 2 Report

1. What is the main question addressed by the research?
2. Do you consider the topic original or relevant in the field?  Yes

3. Does it address a specific gap in the field? Yes

4. What does it add to the subject area compared with other published material?

A good analysis of the oxidation behavior of Nb-Si high temperature alloys.

5. What specific improvements should the authors consider regarding the methodology?

Describe further the experimental section. Furnace, devices and measurements. A porosity quantification and analysis is required. 

6.What further controls should be considered?

Describe the way in which the volume of phases was calculated.

7. Are the conclusions consistent with the evidence and arguments presented and do they address the main question posed? Yes

8. Are the references appropriate? Yes

9. Please include any additional comments on the tables and figures.

Error bars are missing in Tables 3 and 4.

I think that the explanation of the oxidation behavior can be improved, I mean, this paragraph:

This indicates that the sintering reaction of the powders is 361 not sufficient at such lower hot press sintering temperature, resulting in the alloy being 362 not dense, thus its high temperature oxidation resistance is poor. When the hot press sin-363 tering temperature was higher, the sintered alloys have nearly equiaxed grains. The sin-364 tering reaction is much more sufficient. The reaction diffusion of each element in the alloys 365 is also sufficient, resulting in more compact microstructure. Therefore, the high-tempera-366 ture oxidation resistance of the alloys is obviously improved.

Discussion can be complemented and talk about the free surface area of the particles that did not fully react. I mean Ti, Nb, Cr in the powder form or sponge-like where diffusion of oxygen takes place. (Fig. 11 a, b and c show a porous microstructure due to the incomplete sintering process, then those are more prone to oxidation). 

Fig. 11 images are really good, a good complement to Fig. 2. Are those images representative of the microstructure? Because, Fig. 2 is too general with low magnification.

There are only some minor issues with the English about style and redaction. It is not necessary to make corrections.

Reviewer 3 Report

In this paper, mechanical properties and oxidation resistance of Nb-Si based ultrahigh temperature alloys prepared by hot press sintering were investigated. I think there are unclear points and misunderstandings in this manuscript.

(1)   Authors claimed that SiO2 existed in amorphous form, and its content in the oxide film is relatively low. Why does only SiO2 form in an amorphous state? In addition, it is difficult to know the volume fraction of SiO2 because there is no low-magnification BSE image.

(2)   It is hard to understand the structure of the oxide film of this alloy. Authors should indicate low-magnification BSE images and illustrate a schematic diagram of the oxide films.

(3)  Authors show Gibbs free standard energy as evidence of the preferential formation of TiO2. However, thermodynamics calculation can only be used if the oxide formation is stable or not. Authors can only claim that Gibbs standard formation energies of TiO2 and Nb2O5 are both negative, then both could form.

Round 2

Reviewer 1 Report

The authors corrected the submitted manuscript professionally.